# TeRF: Text-driven and Region-aware Flexible Visible and Infrared Image Fusion

Submission Id: 2025

## ABSTRACT

The fusion of visible and infrared images aims to produce high-quality fusion images with rich textures and salient target information. Existing methods lack interactivity and flexibility in the execution of fusion. It is unfeasible to express the requirements to modify the fusion effect, and the different regions in the source images are treated equally across the identical fusion model, which causes the fusion homogenization and low distinction. Besides, their pre-defined fusion strategies invariably lead to monotonous effects, which are insufficiently comprehensive. They fail to adequately consider data credibility, scene illumination, and noise degradation inherent in the source information. To address these issues, we propose the **Te**xt-driven and **R**egion-aware **F**lexible visible and infrared image fusion, termed as TeRF. On the one hand, we propose a flexible image fusion framework with multiple large language and vision models, which facilitates the visual-text interaction. On the other hand, we aggregate comprehensive fine-tuning paradigms for the different fusion requirements to build a unified fine-tuning pipeline. It allows the linguistic selection of the regions and effects, yielding visually appealing fusion outcomes. Extensive experiments demonstrate the competitiveness of our method both qualitatively and quantitatively compared to existing state-of-the-art methods.

## CCS CONCEPTS

• **Computing methodologies → Computer vision**.

## KEYWORDS

Image fusion, text-driven, large models, fine-tuning

## 1 INTRODUCTION

Existing imaging technologies rely on specialized optical systems to capture and convert radiation information, which generates optical images with distinct characteristics. However, these images often provide a limited and partial modality of the diverse information presenting within the identical scene, thereby failing to adequately describe the entire scene. In this context, the image fusion technique has come into being with the objective of extracting pertinent and complementary information from diverse source images and amalgamating it into fused images. In this field, visible and infrared images fusion (VIF) stands out as one of the most widely adopted

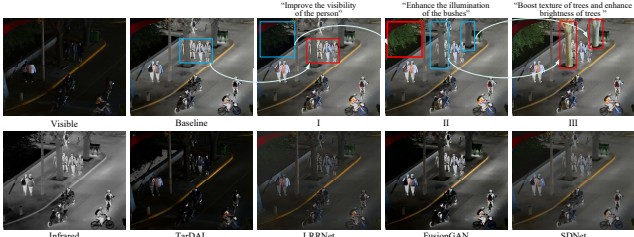

**Figure 1: Visual comparisons of VIF between TeRF and four state-of-the-art methods on the LLVIP dataset.**

techniques [33, 35]. Visible images typically capture intricate spatial details and plentiful color in the scene, but are susceptible to environmental lighting conditions. Infrared images record the thermal radiation information to reflect the salient characteristics of targets, but barely carry scene textures and colors. Hence, VIF aims to generate high-quality images with pronounced saliency and intricate texture details, as depicted in Figure 1. Harnessing these advantages, VIF technology holds promise for various applications, including target detection [18], military surveillance [3], autonomous driving [22], and others [16].

To date, the process of visible and infrared image fusion can be delineated into three key stages: feature extraction, feature fusion, and image reconstruction. Initially, it is imperative to extract representative features from the source images by employing specific transformations or projections. Subsequently, the complementary features are seamlessly integrated by adopting appropriate fusion strategies. Finally, corresponding reverse projections are employed to transform the fused feature back to the common image domain, thereby facilitating the process of image reconstruction. Generally, traditional VIF methods rely on manually designed fusion strategies, which can be further divided into multi-scale transformation-based methods, sparse representation-based methods, subspace-based methods, saliency-based methods, and others, according to the applied transformation [32, 34]. However, these methods often fall short in addressing diverse working conditions, as their fusion performance is constrained by limited linear fitting capabilities.

Over the past few years, deep learning has facilitated leaps and bounds in the visible and infrared fusion task owing to its powerful non-linear fitting ability. According to the structures of deep models, deep-learning-based methods can be categorized into convolution neural network (CNN)-based methods, auto-encoder (AE)-based methods, generative adversarial network (GAN)-based methods, transformer-based methods, and others. CNN-based methods [9, 14] leverage the remarkable local connectivity characteristics to explore the hierarchical patterns and features, which promotes the integration of complementary information to synthesize high-quality fused images. AE-based methods [36, 40] typically contain a two-terminal

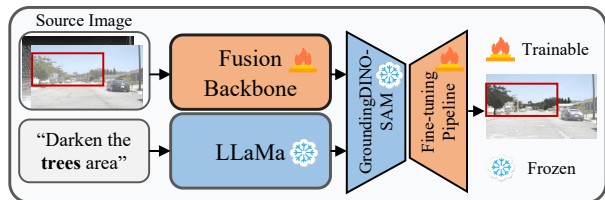

Figure 2: Overview of our proposed TeRF.

structure, where the front encoder seeks the efficient representation of the source images and the end decoder restores them with minimal loss. After that, the encoded representations of different source images are combined under the predefined fusion strategy before generating fused images through the decoder. GAN-based methods [15, 19, 23] utilize multiple generators and discriminators to explore the feature distributions of paired source images and perform appropriate constraints to manipulate the style and the content of the fused images. Transformer-based methods [17, 24, 27, 37] incorporate a novel attention mechanism that captures the long-range dependencies to focus on relevant parts of the source information while downplaying the irrelevant information. This effectively improves network efficiency and enhances convergence performance.

Despite the effectiveness of deep-learning-based methods in integrating complementary information from heterogeneous modalities of visible and infrared images [11, 28, 43], there still exist several limitations. On the one hand, interactivity and flexibility are absent in the execution of fusion. It is unavailable to express the requirements to modify the corresponding fusion effects since the models lack a command interface. Besides, existing methods uniformly process various regions of the source images, which leads to fusion homogenization and low distinction across the identical model. Essentially, it retains the generality but sacrifices the diversity of fusion results. On the other hand, the predefined fusion strategies are insufficiently comprehensive in the credibility of source information, refinement of lighting conditions, and noise removal. Specifically, current models struggle to effectively allocate the varying degrees of fusion tendency for different source images, presenting a singularity that cannot be applied to all scenarios. In addition, they lack adaptability in handling inappropriate light conditions, where texture and color fidelity deteriorate within the scene. Moreover, noise is inadequately taken into account as it inevitably exists and reduces the image quality. These issues lead to poor information reliability, weak information discovery ability, and low information fidelity, which seriously limits the scope of application and performance of existing fusion methods [7, 14, 18, 41].

To address these challenges, we integrate the advantages of pretrained large vision models, large language models, and multiple fine-tuning paradigms to achieve **Te**xt-driven and **R**egion-aware **F**lexible visible and infrared image fusion, termed **TeRF**, as shown in Figure 2. For interactivity and flexibility, we introduce the language model LLaMa [25] as a text analyzer, which leverages the in-context learning to accurately recognize the relevant prompts for segmentation targets and fine-tuning tasks. After that, we combine the vision models to achieve region-aware effects, where the GroundingDINO [13] and SAM [5] models cooperate for the fine-grained semantic segmentation in a successive manner. Benefiting

from the priors of the large models, text-driven interaction and region-aware flexibility can be effectively achieved. For the incompleteness of fusion strategies, a high-performance fusion backbone and a unified fine-tuning pipeline are devised. The former provides high-quality fusion results, serving as precursors with enhanced visual effects and improved detection performance. The latter leverages comprehensive fine-tuning paradigms to fully facilitate multi-task feasibility, which comprises the modification of fusion tendency, the adjustment of illumination, and the removal of noise. This effectively improves the completeness of the fusion strategies and the practicality in various scenarios. As illustrated in Figure 1, our basic performance outperforms the representative methods, and we can linguistically modify the fusion effects on specified regions in a continuous manner(I-III). Our contributions are summarized below:

- We propose a text-driven and region-aware framework for VIF with high interactivity and flexibility. It ensembles the large language and vision models for linguistically modifying the fusion effects of different regions.
- A high-performance fusion backbone is devised to attain superior fusion precursors. Furthermore, a unified fine-tuning pipeline is constructed for the flexible fusion modification, which fully concerns the comprehensive fusion strategies.
- Extensive experiments are conducted to demonstrate the superiority of our method in terms of interactivity, flexibility, and completeness both qualitatively and quantitatively.

## 2 METHODOLOGY

For convenience and generality, the used notations are summarized here. Two heterologous source images including infrared image and visible image are denoted as $I_{ir}$ and $I_{vi}$, respectively. The fused image generated by the pretrained fusion network $\mathcal{F}(\cdot)$ is denoted as $I_f$, while the image acquired by fine-tuning pipeline $\mathcal{T}(\cdot)$ is defined as $I_S$. The employed LLaMa for text parser is determined as $\Psi(\cdot)$. The applied vision model for text-assisted region segmentation is defined as $\mathcal{R}(\cdot)$, which consists of GroundingDINO and SAM.

### 2.1 Overview of The Framework

The realization of interactivity and flexibility within TeRF is illustrated in Figure 3. To start with, we define the instruction prompts with a few examples and demonstrations, thereby fine-tuning the LLaMa to accommodate entity recognition and text classification. After that, the predefined instruction prompts as well as input text commands $\boldsymbol{\varphi}$ are combined together to form the contextual information, which enables the large language model to generate the output in the specific format from the provided demonstration in the in-context learning manner. Subsequently, the segmentation prompts $\boldsymbol{\varphi}_{seg}$ and fine-tuning prompts $\boldsymbol{\varphi}_{task}$ are split and simplified, which can be formulated as:

$$\langle \boldsymbol{\varphi}_{seg}, \boldsymbol{\varphi}_{task} \rangle = \Psi(\boldsymbol{\varphi}). \tag{1}$$

Meanwhile, the fusion of visible and infrared images is also being implemented. The pre-trained fusion model is devoted to synthesizing the fused result $I_f$ by integrating salient target information and rich texture information from the source images. In this way, the fused image potentially yields enhanced visual effects and improved

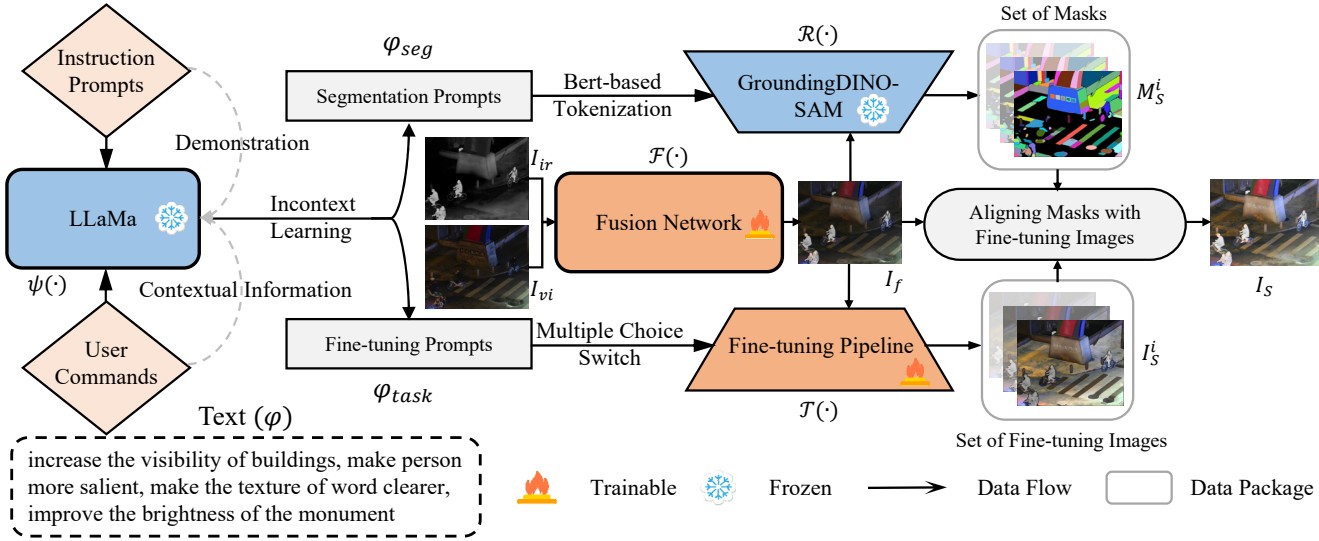

**Figure 3: The detailed framework of TeRF.**

detection performance. This process can be defined as:

$$I_f = \mathcal{F}(I_{ir}, I_{vi}; \theta_{\mathcal{F}}), \tag{2}$$

where $\theta_{\mathcal{F}}$ is the pretrained parameters of fusion network. Then the simplified prompts as well as fused images are served for both large vision models and fine-tuning pipeline. $\boldsymbol{\varphi}_{seg}$ is embedding into token by predefined bert-based encoder [2] at first. Subsequently, it assists the pre-trained vision models in locating the target and segmenting the pertinent regions, ultimately generating a set of text-related segmentation masks. Notably, this process is in a successive manner. At first, GroundingDINO simultaneously receives textual and visual signals, leveraging them to generate bounding boxes as visual cues. These cues provide a coarse localization of the target, facilitating subsequent fine-grained segmentation by the SAM model. Given that SAM lacks an interface for direct text-to-semantic segmentation, such a cascaded approach becomes necessary. The entire process can be denoted as:

$$M_S = \mathcal{R}(I_f, \boldsymbol{\varphi}_{seg}), \tag{3}$$

where $M_S$ stands for the collected semantic masks. At the same time, $\boldsymbol{\varphi}_{task}$ triggers the switch of the fine-tuning pipeline, where the task-related fine-tuning paradigm is activated for the modification of the fusion effect. It can be described as:

$$I_S^i = \mathcal{T}(I_{ir}, I_{vi}, I_f, \boldsymbol{\varphi}_{task}; \theta_S^i), \tag{4}$$

where $I_S^i$ represents the $i^{th}$ fine-tuning image and $\theta_S^i$ are the learnable parameters for the specific tasks. At last, a set of fine-tuning images $I_S$ along with the set of corresponding masks are aligned to merge the final outputs, as formulated below:

$$I_S = \sum_{i=1}^{N} M_S^i \odot I_S^i + (1 - M_S^i) \odot I_f, \tag{5}$$

where $M_S^i$ denotes the $i^{th}$ masks, $N$ represents the total number of masks, and the symbol $\odot$ stands for the element-wise addition.

## 2.2 Structure Designs

As aforementioned, two major parts of designed networks are the predefined fusion network and fine-tuning pipeline.

The former network designs are illustrated in Figure 4. The infrared image $I_{ir}$ and visible image $I_{vi}$ are concatenated together and sent to the feature extraction module at first. It contains a convolutional layer with $PReLU(\cdot)$ as activation in the front and a two-layer convolutional layer with $PReLU(\cdot)$ in the end. Subsequently, three residual dense blocks are employed to perform hierarchical feature exploration. These features are then sent to the feature refinement module, which comprises three consecutive convolutional layers and a $ReLU(\cdot)$ activation function at the end. After that, the preceding features are expended in the channel dimension, which are packed into transformer-based attention blocks to enhance the relevant features and weaken the others. At last, features flow into the feature integration module, which contains a front convolutional layer, a pixel shuffle layer [10], and an end convolutional layer.

The latter network designs are represented in Figure 5. The basic element of the fine-tuning pipeline is the lightweight fine-tuning unit, which is constituted by three repeated convolutional layers with a $ReLU(\cdot)$ activation at the head and a final convolutional layer at the tail. Depending on different fine-tuning tasks, a variable number of units are employed. In cases of a tendency bias towards the fusion effect, it suffices to unlock the predefined fusion network and perform minor optimization. In cases of illumination adjustment, three fine-tuning units are employed to improve the quality of exposure. As for denoising tasks, a single fine-tuning unit is introduced for noise estimation and removal.

## 2.3 Loss Functions

The purpose of VIF is to retain the ample spatial information and the salient information of the source images. Hence, we strip away the spatial and color attributes from RGB space to YCrCb space. Accordingly, we preserve the maximum intensity and gradient of

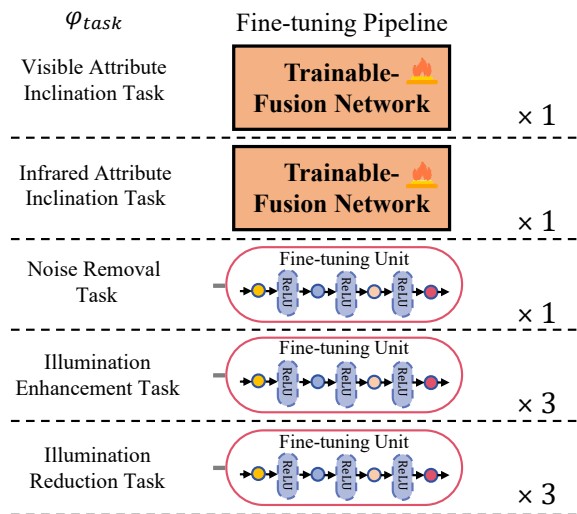

Figure 4: The structures of the fusion network.

Figure 5: The architecture of the fine-tuning pipeline.

the source images on the $Y$ channel in order to fulfill the purpose, which can be expressed as below:

$$\mathcal{L}_Y = \|Y_f - \max(Y_{ir}, Y_{vi})\|_1 + \|\nabla Y_f - \max(\nabla Y_{ir}, \nabla Y_{vi})\|_1, \quad (6)$$

where $Y_{(\cdot)}$ is the Y-channel of each image, $\nabla$ stands for the Sobel operation, and $\|\cdot\|_1$ represents $l_1$-norm. To ensure minimal chromatic aberration in the fused images, we constrain them to align closely with visible images in the other two channels. It can be formulated as:

$$\mathcal{L}_{\text{color}} = \|Cr_f - Cr_{vi}\|_1 + \|Cb_f - Cb_{vi}\|_1, \quad (7)$$

where $Cr_{(\cdot)}$ and $Cb_{(\cdot)}$ are the Cr-channel and Cb-channel of the corresponding source image, respectively. Besides, we devise a structural consistency loss to constrain the structures of fused images and source images to be similar, which can be defined as:

$$\mathcal{L}_{\text{ssim}} = (1 - \text{SSIM}(I_f, I_{vi})) + (1 - \text{SSIM}(Y_f, Y_{ir})), \quad (8)$$

where $\text{SSIM}(\cdot)$ computes the structural similarity of the image pair. Concerning the above three points, the overall loss functions can be formulated as below:

$$\mathcal{L}_{\text{fuse}} = \alpha \mathcal{L}_Y + \beta \mathcal{L}_{\text{color}} + \eta \mathcal{L}_{\text{ssim}}, \quad (9)$$

where $\alpha$, $\beta$ and $\eta$ are the trade-off hyper-parameters.

## 2.4 Multi-task Accessibility

*2.4.1 Visible or Infrared Attribute Inclination.* VIF can be viewed as a generative task without departing from the source information. Hence, the variety of the fusion effect is inseparable from the constraints of the source information. The fused and source images are packed into the updated fusion models to execute the fusion tendency by reformulating the loss function. To preserve the spatial details, gradient-related losses remain unchanged, while modulation is performed through intensity and structure-related terms. For the inclination of visible attribute, $\mathcal{L}_Y$ and $\mathcal{L}_{\text{ssim}}$ are redefined as $\mathcal{L}_Y^{vi*}$ and $\mathcal{L}_{\text{ssim}}^{vi*}$:

$$\mathcal{L}_Y^{vi*} = \|Y_f - Y_{vi}\|_1 + \|\nabla Y_f - \max(\nabla Y_{ir}, \nabla Y_{vi})\|_1, \quad (10)$$

$$\mathcal{L}_{\text{ssim}}^{vi*} = (1 - \text{SSIM}(I_f, I_{vi})). \quad (11)$$

Correspondingly, for the inclination of the infrared attribute, they are redefined as $\mathcal{L}_Y^{ir*}$ and $\mathcal{L}_{\text{ssim}}^{ir*}$:

$$\mathcal{L}_Y^{ir*} = \|Y_f - Y_{ir}\|_1 + \|\nabla Y_f - \max(\nabla Y_{ir}, \nabla Y_{vi})\|_1, \quad (12)$$

$$\mathcal{L}_{\text{ssim}}^{ir*} = (1 - \text{SSIM}(Y_f, Y_{ir})). \quad (13)$$

*2.4.2 Enhance or Weaken The Illumination.* Following the classical Retinex theory [6], an image can be decomposed into the product of a reflectance map $R$ and an illuminance map $L$, with the addition of noise $N$. Among them, $R$ describes the intrinsic property of the captured scene, and $L$ represents the illumination intensity emitted by the light source. Consequently, within the framework of RRDNet [42], we employ three independent fine-tuning units to extract these components and appropriately adjust the illuminance map. The devised loss can be formulated as below:

$$\mathcal{L}_r = \|I_f - (R \odot S + N)\|_1 + \|S - S_0\|_1 + \|R - \frac{I}{S}\|_1, \quad (14)$$

$$\mathcal{L}_t = \|\frac{(\nabla_x S)^2}{G \circ ((\nabla_x Y_f)^2)}\|_1 + \|\frac{(\nabla_y S)^2}{G \circ ((\nabla_y Y_f)^2)}\|_1, \quad (15)$$

$$\mathcal{L}_n = \|I_f \odot N\|_F + \frac{1}{\lambda_n}\left[\|w_n(\nabla_x R)^2\|_1 + \|w_n(\nabla_y R)^2\|_1\right], \quad (16)$$

where $S_0$ is the maximum value of the RGB channel, $G \circ (\cdot)$ is the gaussian filter. $\nabla_x$ and $\nabla_y$ are the Sobel operators in the horizon and vertical directions, respectively. $w_n$ equals normalization of $1/(I_f \odot (\nabla_x R)^2 \odot (\nabla_y R)^2)$. The total loss is denoted as:

$$\mathcal{L}_{\text{lumin}} = \mathcal{L}_r + \lambda_t \mathcal{L}_t + \lambda_n \mathcal{L}_n, \quad (17)$$

where $\lambda_t$, $\lambda_r$ and $\lambda_n$ are the trade-off parameters. After the extraction of the key components, we perform gamma correction on the illumination map for the reconstruction of the enhanced image.

*2.4.3 Noise Removal.* As the pretrained fusion model cannot provide denoising priors, a fine-tuning unit is introduced additionally. Following the framework of ZSN2N [21], the unit is optimized using the following loss functions to achieve noise removal:

$$\mathcal{L}_{\text{res}} = \|D_1(x) - f_\theta(D_1(x)) - D_2(x)\|_2^2 + \tag{18}$$
$$\|D_2(x) - f_\theta(D_2(x)) - D_1(x)\|_2^2,$$

$$\mathcal{L}_{\text{cons}} = \|D_1(x) - f_\theta(D_1(x)) - D_1(x - f_\theta(x))\|_2^2 \tag{19}$$
$$+ \|D_2(x) - f_\theta(D_2(x)) - D_2(x - f_\theta(x))\|_2^2,$$

where $x$ is a noise image, $D_1$ and $D_2$ are the manually designed kernels for down-sample operation to derive the paired noise image, $\theta$ is the parameters of the fine-tuning unit $f$. The total loss functions $\mathcal{L}_{N2N}$ can be expressed as:

$$\mathcal{L}_{N2N} = \mathcal{L}_{\text{res}} + \mathcal{L}_{\text{cons}}, \tag{20}$$

where $\mathcal{L}_{\text{res}}$ yields the residual information constraints and $\mathcal{L}_{\text{cons}}$ provides the consistency constraints for better performance as well as avoiding the over-fitting.

*2.4.4 Repeatability and Continuity for Fine-tuning.* When the text instructions involve different requirements for effects or regions, the fine-tuning units are reinitialized and optimized accordingly. However, when it is required to conduct identical fine-tuning tasks for the same region consecutively, it is imperative to guarantee that the effects get better as the increase of fine-tuning iterations. Regarding the inclination of the source attribute, the fusion network reloads the previous model updated in the last fine-tuning iteration and continues optimization for the fixed epochs. For illumination adjustment, as the reflectance map of the image is independent and stays unchanged, It simply necessitates performing corresponding transformations on the illumination map multiple times. Concerning the noise removal, the original noise prior remains unchanged, but more optimization epochs are performed to continuously improve the performance of the reloaded model. In summary, comprehensive fusion strategies are fully developed.

# 3 EXPERIMENTAL RESULTS AND ANALYSIS

## 3.1 Experiment Settings

*3.1.1 Datasets and Benchmark.* We conduct fusion performance validation on three different datasets, namely RoadScene, MSRS, and LLVIP. RoadScene datasets contain a total of 221 visible and infrared image pairs. We randomly allocate 167 pairs for training and validation and 54 pairs for testing. MSRS datasets are officially divided into 1803 image pairs for training and 361 image pairs for testing with day-time and night-time labels. LLVIP datasets include a large number of source image pairs, most of which are captured at different time with fixed camera angles. We keep the default 12025 pairs of source images as the training set and select 136 image pairs with significant content differences from the given 3463 pairs of test images as the test set. Our method is compared with eight deep-learning-based state-of-the-art fusion algorithms, *i.e,* DDFM [38],

FusionGAN [20], DIDFuse [39], Res2Fusion [29], RFNNest [8], SD-Net [31], LRRNet [9], TarDAL [12]. For quantitative assessment, five metrics are selected to objectively evaluate the fusion performance, including average gradient (AG) [1], structural similarity index (SSIM) [26], visual information fidelity (VIF) [4], mutual information (MI), and gradient-based quality index $Q_{AB/F}$ [30].

*3.1.2 Training Configuration.* TeRF comprises two trainable parts, including a fusion network and a fine-tuning pipeline. During the training process of the fusion network, the total training epoch is set as 100 and the batch size is set as $b = 32$. The learning rate is $3 \times 10^{-4}$ with the default Adam optimizer. The hyper-parameters are fixed as follows: $\alpha = 1, \beta = 1, \eta = 0.1$. Regarding the fine-tuning pipeline, the inclination of the visible or infrared attributes is conducted by the meticulously regulating fusion network. The learning rate is reduced to $8 \times 10^{-5}$ to optimize for 50 epochs per time. As for the illumination adjustment, trade-off parameters of the loss functions are defined: $\lambda_t = 1, \lambda_n = 5000$. The learning rate is set as $5 \times 10^{-4}$ and the fine-tuning epochs are set to 500. Besides, the learning rate for noise removal is $1 \times 10^{-3}$ with optimization for 100 epochs per time. All the parameters of the fine-tuning units are initialized by Xavier initialization and are equipped with the default Adam optimizer for optimization. All the experiments are conducted on a desktop with a 2.6GHz AMD EPYC 7H12 CPU, NVIDIA GeForce RTX 3090, and 128 GB RAM memory.

## 3.2 Comparative Experiment

*3.2.1 Qualitative Comparisons.* The visual comparisons over three datasets are depicted in Figure 6. Evidently, FusionGAN emphasizes brightness overly at the expense of fine details. RFNNest fails to preserve sufficient spatial information and yields ambiguous effects. LRRNet and SDNet effectively retain the texture details albeit at the expense of color fidelity, which leads to diminished luminance. TarDAL and DDFM maintain the balance of color information and salient information better, but the contrast of the fusion result is not high enough. In particular, when visible information and infrared information are completely opposite, the fusion strategy adopted by DIDFuse leads to information cancellation and distortion. Res2Fusion insufficiently integrates the infrared information, which reduces the saliency of targets. In contrast, our method effectively retains the ample textures within the scenes while rationally integrating salient information from the source images. Moreover, the applied fine-tuning paradigms further enhance the visual effect of the fusion results with the assistance of texts, which showcases flexibility and interactivity. Consequently, our method achieves favorable results and outperforms others.

*3.2.2 Quantitative Assessment.* To further validate the competitiveness of our approach, we conduct the quantitative evaluation of five metrics. For fairness, we exclusively compare the performance of our baseline with other methods, and the impacts of fine-tuning tasks are individually discussed in the next section. The quantitative results are reported over three datasets in Table 1. Upon analysis of the results, our basic fusion model achieves the top scores of all five metrics on both MSRS and LLVIP datasets. It indicates that our method retains rich gradient information and maintains the highest fidelity of source information. However, our baseline attains the

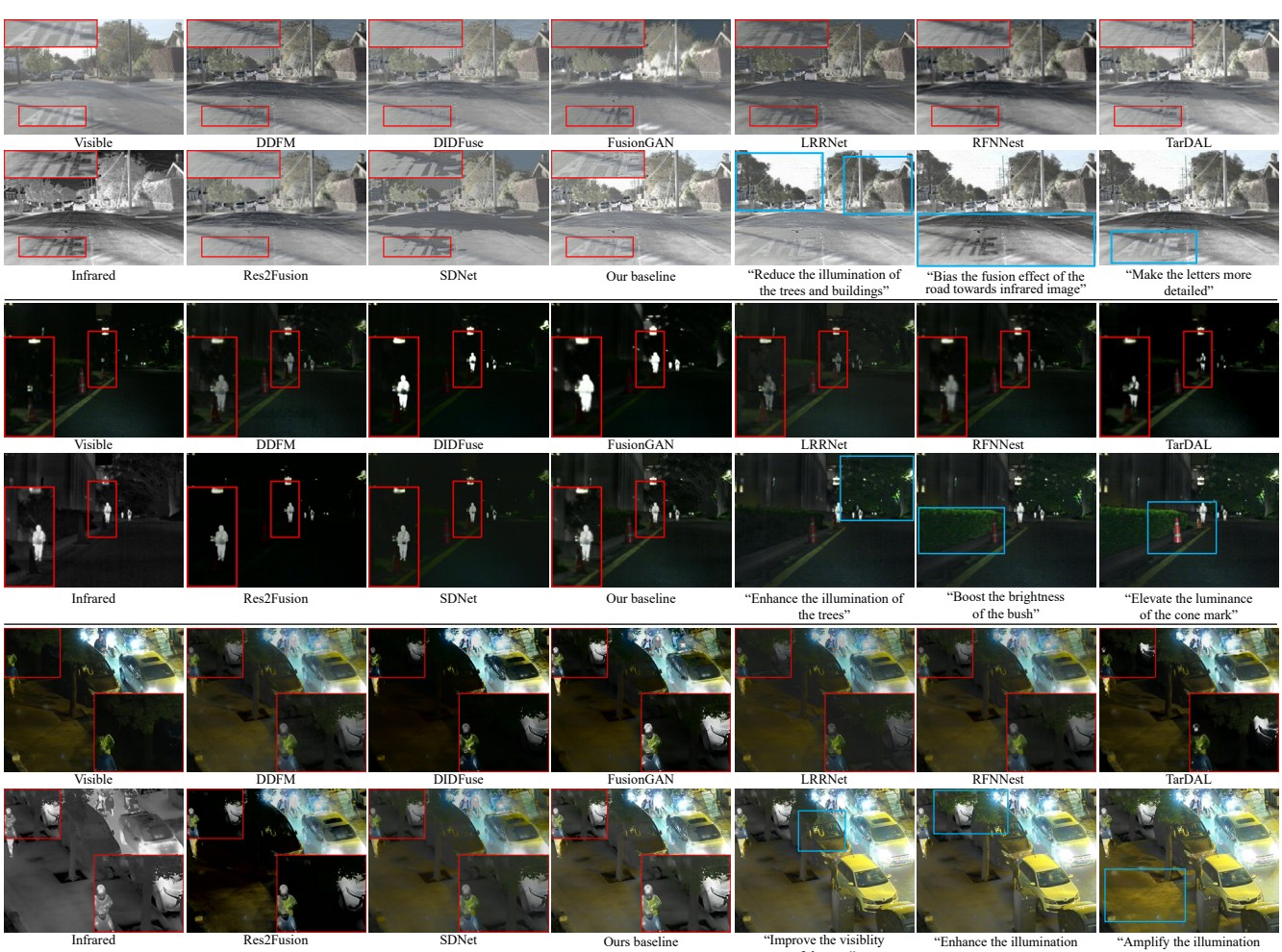

Figure 6: Qualitative comparison of RoadScene, MSRS, and LLVIP datasets from top to bottom, respectively.

Table 1: The quantitative results over three datasets. The best and the second best values are highlighted by bold and underline, respectively. The up arrows indicate higher values correspond to better results.

| Method | RoadScene | | | | | MSRS | | | | | LLVIP | | | | |
|---|---|---|---|---|---|---|---|---|---|---|---|---|---|---|---|
| | AG↑ | SSIM ↑ | VIF↑ | MI ↑ | $Q_{AB/F}$↑ | AG↑ | SSIM ↑ | VIF↑ | MI ↑ | $Q_{AB/F}$↑ | AG↑ | SSIM ↑ | VIF↑ | MI ↑ | $Q_{AB/F}$↑ |
| DDFM | 4.3445 | 0.6905 | 0.6349 | 2.1311 | 0.5003 | 2.7070 | 0.7088 | 0.7420 | 1.8423 | 0.4729 | 2.3232 | 0.6475 | 0.5489 | 1.7080 | 0.2196 |
| FusionGAN | 3.4935 | 0.6173 | 0.2809 | 1.9817 | 0.2570 | 2.4327 | 0.5575 | 0.3335 | 1.2951 | 0.1822 | 4.3900 | 0.5276 | 0.5141 | 1.9021 | 0.3790 |
| DIDFuse | 3.7352 | 0.7262 | 0.5992 | 2.1793 | 0.4389 | 2.7800 | 0.6179 | 0.4197 | 1.5337 | 0.2360 | 2.6855 | 0.4472 | 0.4428 | 1.6878 | 0.3038 |
| Res2Fusion | 4.0473 | 0.6410 | 0.5800 | 2.0653 | 0.4283 | 2.3889 | 0.3622 | 0.3941 | 1.5902 | 0.3477 | 2.8024 | 0.3545 | 0.3322 | 0.9078 | 0.2381 |
| SDNet | 3.8387 | 0.6822 | 0.4474 | 2.2649 | 0.3774 | 2.2093 | 0.6530 | 0.4318 | 1.9996 | 0.2338 | 2.8215 | 0.6517 | 0.5750 | 2.2802 | 0.3400 |
| RFNNest | 3.2002 | 0.6466 | 0.5421 | 2.0148 | 0.3159 | 2.3240 | 0.6885 | 0.6547 | 1.6982 | 0.3899 | 2.3802 | 0.6667 | 0.6166 | 1.7312 | 0.2791 |
| TarDAL | 3.9925 | 0.6868 | 0.5597 | 2.3469 | 0.3993 | 3.2730 | 0.5093 | 0.6728 | 1.8340 | 0.4255 | 2.0986 | 0.5911 | 0.5422 | 1.6904 | 0.2155 |
| LRRNet | 4.0987 | 0.5675 | 0.3203 | 1.9262 | 0.2625 | 2.8547 | 0.5954 | 0.5413 | 2.0256 | 0.4545 | 2.8259 | 0.6278 | 0.5319 | 1.5831 | 0.4091 |
| Ours | 5.1801 | 0.7041 | 0.6678 | 2.5579 | 0.5912 | 3.4806 | 0.7114 | 0.9472 | 2.9170 | 0.6665 | 4.3990 | 0.6775 | 0.9429 | 3.0141 | 0.7006 |

best performance across AF, VIF, MI, and $Q_{AB/F}$ but ranks second in SSIM on RoadScene datasets. This may be attributed to the significant disparity between visible and infrared image information, making it difficult for the fused image to simultaneously accommodate the structural similarities of both. In general, our method surpasses others and is the most competitive.

## 3.3 Impact of Fine-tuning Tasks

We conducted a series of experiments to assess the impacts of fine-tuning tasks on the RoadScene datasets. The visual comparisons are illustrated in Figure 7, while the metrics evaluation is presented in Table 2. Each task is performed once across the entire region

**Table 2: Quantitative results of the different fine-tuning tasks on the RoadScene dataset.**

| Category | AG↑ | SSIM ↑ | SF ↑ | $Q_{AB/F}$↑ | EN ↑ | SD ↑ |
|---|---|---|---|---|---|---|
| Baseline | 5.1801 | 0.7041 | 7.5797 | 0.5912 | 7.1221 | 14.5694 |
| (I) | 5.2003 | 0.6739 | 7.5821 | 0.6191 | 7.5234 | 18.2078 |
| (II) | 3.1420 | 0.6743 | 5.6145 | 0.4150 | 6.7444 | 12.5711 |
| (III) | 5.0511 | 0.6515 | 7.5760 | 0.5051 | 6.7977 | 12.3864 |
| (IV) | 5.2512 | 0.7009 | 7.6138 | 0.6245 | 7.3150 | 16.6660 |
| (V) | 8.5599 | 0.4670 | 12.1157 | 0.4023 | 7.2400 | 14.5627 |
| (VI) | 6.6462 | 0.5741 | 10.1196 | 0.4235 | 7.2187 | 14.3155 |
| (VII) | 5.5428 | 0.6596 | 8.3990 | 0.5106 | 7.1489 | 14.3491 |

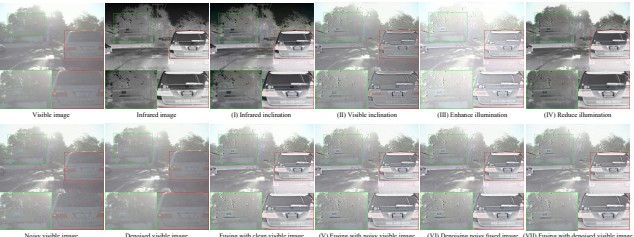

Figure 7: Fine-tuning example on the whole fusion image.

of the source images. (I) and (II) are the inclination of infrared and visible attributes, respectively. (III) and (IV) are on behalf of illumination enhancement and reduction, respectively. Concerning the facts that the fused image lacks the ground-truth as a reference and noise degradation is always imposed on the visible images, we simulated the (V) by fusing with Gaussian noisy visible images. The assessments are performed between them and noise-free source images. (VI) conducts noise removal for the fused images of (V) directly, while (VII) denoises the noisy visible image first and then performs fusion to generate the fused images. Furthermore, we evaluate results on two reference metrics SSIM and $Q_{AB/F}$, and four source-free metrics AG, spatial frequency (SF), entropy (EN), and standard deviation (SD).

As for the inclination of source attributes, the gradient-related indicators (*e.g.* AG, SF, $Q_{AB/F}$) of (I) increase, while these of (II) decrease. It indicates that the brightness changes of infrared images are drastic on this dataset. In addition, EN and SD also keep the same change in (I) and (II). Since fine-tuning approaches are not for neutral fusion effects, the value of SSIM inevitably reduces. For the illumination adjustment, the two fine-tuning methods are completely opposites. All the metrics expect for SSIM, get lower in (III) while higher in (IV). Compared to the baseline, enhancing illumination leads to a decrease in these metrics as most scenes in the datasets may be overexposed. Conversely, reducing illumination leads to a more balanced distribution of information. Similarly, both approaches make the fused information deviate from the source information, inevitably leading to a decrease in SSIM. Regarding noise removal, it is observed that the introduction of noise significantly deteriorates the image quality as the value of most metrics gets worse. However, the increase in AG and SF can be attributed to the alteration of intensity coherence caused by noise. As fusion can

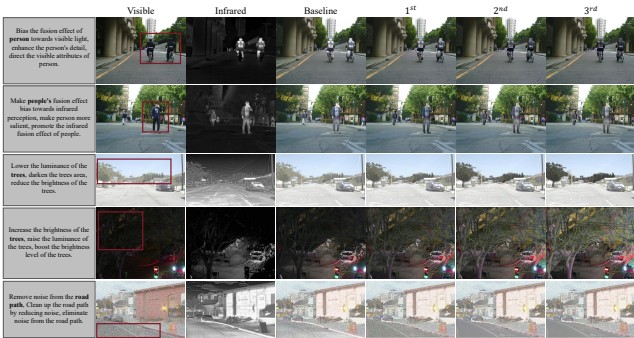

Figure 8: Examples of performing the fine-tuning on the same region consecutively with similar text instructions.

alter the priors of noise originating from the source image, fusing with the denoised source image is better than directly denoising the noisy fused image. Credibly, noise removal can restore most metrics and visual effects.

## 3.4 Text-driven Region-aware Fusion

Due to the excessive number of regions in the image, artificial one-to-one text modulation is not feasible. Therefore, we present qualitative results in Figures 8 and 9 for two different configurations. One involves consecutively employing the same fine-tuning effect for the same region with similar texts, the other is fine-tuning different areas randomly with different texts. It is shown that our approach can accurately analyze segmentation targets and fine-tuning tasks from different text representations. When performing the same fine-tuning tasks repeatedly, it can progressively enhance the effects. In addition, when dealing with multi-objective fine-tuning tasks, it can harmoniously integral different effects of different objectives. In general, the high-quality fine-tuning results verify that TeRF achieves flexible image fusion in handling diverse tasks and targets.

## 3.5 Time Consuming and Parameters

Furthermore, we conduct the efficiency experiments to record test time consumption and model parameters for all methods, as detailed in Table 3. The majority of deep methods exhibit high speed in processing test image samples. However, the diffusion-based method DDFM necessitates an amount of time for optimizing by performing multi-step noise inference. In comparison, our baseline demonstrates superior performance albeit at the expense of parameters and time, which can be attributed to the incorporation of multiple dense connections and the transformer-based attention mechanism. Hence, it is generally acceptable. In addition, we also provide the running time for each fine-tuning task. $F^{\times 0}$ represents the modification of the fusion attributes without fine-tuning unit. $F^{\times 1}$ refers to the noise removal task with single unit and $F^{\times 3}$ employs three units to adjust the illumination. Obviously, the time cost for fine-tuning is relatively high. Because model training and testing are performed as a whole, the running time includes both of them. However, other deep learning methods require extensive pre-training on large datasets before conducting fast testing.

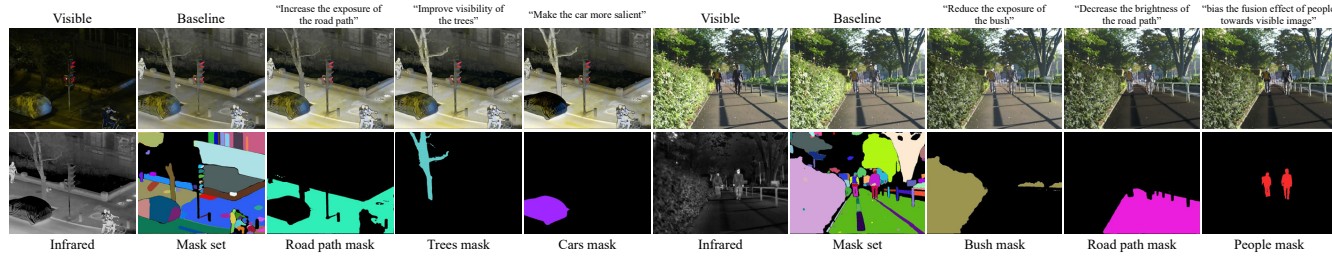

Figure 9: Examples of performing the fine-tuning consecutively with different text instructions.

Table 3: Computational efficiency comparison with eight SOTA fusion methods.

| Method | DDFM | FusionGAN | DIDFuse | Res2Fusion | SDNet | RFNNest | TarDAL | LRRNet | Ours | $F^{\times 0}$ | $F^{\times 1}$ | $F^{\times 3}$ |
|---|---|---|---|---|---|---|---|---|---|---|---|---|
| Time (s) | 49.0303 | 0.0276 | 0.0248 | 0.0378 | 0.0309 | 0.0826 | 0.0578 | 0.0305 | 0.0814 | 26.8641 | 6.2163 | 33.5312 |
| Parameters (M) | 552.663 | 0.926 | 0.260 | 0.098 | 0.064 | 30.097 | 0.296 | 0.196 | 1.765 | 1.765 | 0.0445 | 0.1281 |

Table 4: Detection performance validation on the LLVIP dataset. Bold indicates the best, and underline indicates the second best.

| | Precision | Recall | mAP | | |
|---|---|---|---|---|---|
| | | | @0.50 | @0.75 | @[0.5:0.95] |
| Infrared | 0.823 | 0.762 | 0.854 | 0.566 | 0.534 |
| Visible | 0.771 | 0.626 | 0.723 | 0.340 | 0.391 |
| DDFM | 0.868 | 0.722 | 0.847 | 0.547 | 0.518 |
| FusionGAN | 0.872 | 0.749 | 0.859 | 0.560 | 0.520 |
| DIDFuse | 0.874 | 0.738 | 0.850 | 0.514 | 0.508 |
| Res2Fusion | 0.851 | 0.751 | 0.843 | 0.523 | 0.503 |
| SDNet | 0.860 | 0.755 | 0.853 | 0.573 | 0.529 |
| RFNNest | 0.835 | 0.741 | 0.844 | 0.549 | 0.515 |
| TarDAL | 0.806 | 0.762 | 0.832 | 0.501 | 0.489 |
| LRRNet | 0.860 | 0.757 | 0.857 | 0.556 | 0.522 |
| Baseline | 0.875 | 0.773 | 0.867 | 0.610 | 0.544 |
| Ours* | **0.879** | **0.781** | **0.870** | **0.613** | **0.545** |

## 3.6 Performance on High-level Task

As aforementioned, we leverage the prior within large vision models to achieve region-aware fusion. It may potentially introduce unfairness in comparing the performance of high-level vision tasks using fine-tuning results. Therefore, We compare the detection performance of our baseline with other methods to verify the improvement of downstream fusion performance on high-level tasks. As source attribute inclination alters the characteristics of fused images, the quality of their detection performance largely depends on the original source images. We utilize general text prompts for the illumination adjustments of primary targets and corresponding background to verify priors from large models as well as fine-tuning can promote detection performance, which is denoted as Ours*. We adopted YOLOv8[1] as the object detection backbone and fine-tuning it on the LLVIP datasets. The qualitative comparisons are

[1]https://github.com/ultralytics/ultralytics

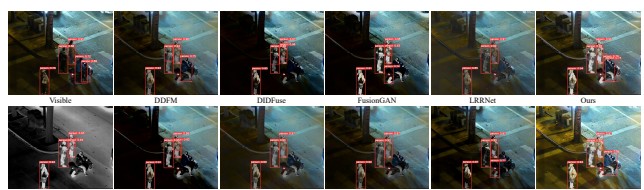

Figure 10: Qualitative comparison of object detection performance on the LLVIP datasets.

displayed in Figure 10. It is obvious that our baseline has higher detection performance, while other methods miss detections or have lower confidence. In addition, the fine-tuning results improve the detection performance as well as visual appeal. Regarding the quantitative assessments, five metrics are provided, including precision rate, recall rate, and mean average precision (mAP) with different threshold values. The results are presented in Table 4. Clearly, our method achieves the highest scores in all five provided metrics, which indicates that our method is the most competitive in terms of performance for object detection. In general, the fine-tuning as well as the priors from the large models improves the detection performance further.

## 4 CONCLUSION

This work proposes a text-driven and region-aware flexible visible and infrared image fusion, termed as TeRF. Multiple large models including LLaMa, GroundingDINO, and SAM, are encompassed to improve the interactivity and flexibility of VIF. Besides, a high-performance fusion network and a fine-tuning pipeline are devised for comprehensive fusion strategies. The former component provides visually appealing fusion precursors with dense connections and the transformer-based attention mechanism. The latter part fully concerns the data credibility, scene illumination, and noise degradation inherent in the source information with unified structures, which improve the information reliability and fidelity. All in all, TeRF allows linguistically segmenting the specific region and modifying the fusion effect accordingly with language instructions.

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
