# OpenReview forum: "TeRF: Text-driven and Region-aware Flexible Visible and Infrared Image Fusion"
_acmmm.org/ACMMM/2024/Conference — MM2024 Poster_

### Official Review · Reviewer_ZQ2b · 2024-04-28

**Rating:** 4
**Confidence:** 3

**Summary:**

This paper tries to use the text information to improve the flexibility of image fusion methods. According to the results, I think this purpose is achieved by employing the llama. The motivation sounds interesting and reasonable, and the performance and visualisations are also good.
However, there are some essential parts are missed. Please refer to the weakness.

**Strengths:**

1. The motivation seems interesting，and also demonstrated effectively by employing the language model.
2. The design is reasonable and from the visualisations, I think it can correctly achieve the purpose..
3. The performance is also good.
4. Most of the experiments are siginificant and can illustrate the purpose.

**Limitations:**

1. The ablation study is missing, which makes the readers confused about the detail contributions of each component.
2. The related works introduced in the Introduction section are not comprehensive, some methods involving the text information into the image fusion task have already been proposed. Such as the TextFusion, which is currently publicly available on Arxiv.
3. The writing should be improved. I can find some mistakes like in line 96, 'feature' should be replaced by 'features', and in fig.2 'Source image' should be 'Source images' since there are two inputs.
4. Achieving controable image fusion is interesting, but it's still less meaningful if the discussion stops here. I suggest the authors to think more about how this helps downstream tasks, which will be more important in my view.

**Suitability:**

3

---

### Official Review · Reviewer_7TXQ · 2024-05-14

**Rating:** 2
**Confidence:** 4

**Summary:**

This manuscript proposes the Text-driven and Region-aware Flexible visible and infrared image fusion, termed as TeRF to address the issues lying in existing methods, such as the lack of interactivity and flexibility. Authors propose a flexible image fusion framework with multiple large language and vision models, which facilitates the visual-text interaction and aggregate comprehensive fine-tuning paradigms for the different fusion requirements to build a unified fine-tuning pipeline. On the whole, the problem mentioned in the paper is interesting, but the novelty of the devised method is weak.

**Strengths:**

1. For interactivity and flexibility, authors introduce the language model LLaMa as a text analyzer, which leverages the in-context learning to accurately recognize the relevant prompts for segmentation targets and fine-tuning tasks. The proposed method combining both text and image information together is suitable for more task-oriented image fusion results.

**Limitations:**

1. Authors define the instruction prompts with a few examples and demonstrations, thereby fine-tuning the LLaMa to accommodate entity recognition and text classification. Since text commands will be input to the LLaMa, the difference between the instruction prompts and the text commands should be clarified. Please give more detailed descriptions of the instruction prompts.
2. The total number of masks N needs a more explicit definition. Does each mask represent one specific object? If so, why the masks shown in Figure 3 are the same? The absence of clear explanation about the mask as well as its settings make readers confused, which significantly reduces the readability of the paper.
3. Experiments to validate the effectiveness of the proposed prompts are insufficient. Ablation study on \phi_seg and \phi_task should be shown to verify the idea of the paper.
4. In Figure 3, there two inputs \phi_task and I_f of the Fine-tuning Pipeline. However in Figure 5, there is only one input \phi_task shown. Please make them consistent.

**Suitability:**

3

---

### Official Review · Reviewer_xizo · 2024-05-20

**Rating:** 5
**Confidence:** 4

**Summary:**

This paper presents a method for fusing visible and infrared images with high flexibility and interactivity, allowing for linguistic modification of the fusion effect in specific areas according to textual instructions. The authors employ the large language model LLaMa to analyze text prompts and utilize large vision models, GroundingDINO and SAM, to identify target regions. Their proposed pipeline incorporates unified fine-tuning to achieve various fusion effects. Overall, the idea of using the large language model and large vision model for image fusion is interesting. Besides, it achieves a relatively good performance along with flexible fusion results.

**Strengths:**

1. The idea of applying multiple large models to assist the fusion process is interesting, as well as the use of text prompts in image fusion. The priors within the large models are fully explored to achieve flexible and interactive fusion.
2. This work addresses practical issues in the fusion of infrared and visible images, including the inability to specify targets/areas for customized fusion, incomplete fusion strategies, and the limitation of achieving a diverse range of fusion effects.
3. It is easy to follow and understand. The experimental results support the effectiveness of the presented method and show relatively good fusion outcomes.

**Limitations:**

1. A comparison of the generalization performance of the fusion model is not provided. It would be better to carry out the experiment on other datasets, (e.g. FMB dataset [1]).
2. In Tab. 3, are the parameters in the large models being computed? And it would be better to provide the Flops.
3. The implementation details are absent when analyzing text prompts. For example, instruction prompts seem to play a big role in LLaMa, but the related information is lacking.
4. Comparing the power of the GroundingDINO and the YOLOv8 performance will be good. It can provide more inspiration for multimodal image fusion.
5. There are some recent works about multi-modality image fusion that have not been surveyed. Their information is as [2-4].

[1] "Multi-interactive feature learning and a full-time multi-modality benchmark for image fusion and segmentation." ICCV, 2023.

[2] "Unsupervised misaligned infrared and visible image fusion via cross-modality image generation and registration." IJCAI, 2022.

[3] "An interactively reinforced paradigm for joint infrared-visible image fusion and saliency object detection." Information Fusion, 2023.

[4] "ReCoNet: Recurrent Correction Network for Fast and Efficient Multi-modality Image Fusion." ECCV, 2022.

**Suitability:**

3

---

### Official Review · Reviewer_6MLM · 2024-05-23

**Rating:** 5
**Confidence:** 4

**Summary:**

A new method combining large language and visual models is designed to perform visible and infrared image fusion of specific areas with text prompts. It considers a complete fusion strategy of data credibility, scene lighting and noise degradation, and builds a unified fine-tuning pipeline. Extensive experiments have demonstrated the advantages of the proposed method over the latest SOTA methods.

**Strengths:**

+ The paper presents a competitive VIF network with high performance that surpasses existing SOTA methods.
+ The paper combines multiple large models and coordinates the processing of multi-modal data, making the fusion effect and region linguistically controllable.
+ The overall fine-tuning framework is flexible, interactive and easy to expand, and has the potential to synthesize text-fusion image pairs.
+ The paper is easy to follow. The motivations and problem statement are explained well.
+ The work can be reproduced based on the provided code.

**Limitations:**

- It is not clear whether the fine-tuning pipeline contains the fusion network or not. In Figure 3 and Figure 5, it has some conflicts.
- More details about the evaluation metrics are helpful for the readers to properly understand the analysis.
- The paper uses three datasets for experiments, but does not cite related articles.
- Since the values of the same metrics are quite close, it might be helpful for the readers if the authors provide also the variance or a statistical test to prove that their method is truly better.
- In the experiment on high-level task, it is necessary to add the baseline experimental results, because the introduction of text may lead to unfair comparison.
- The font size in some of the figures, such as Figure 7 and Figure 10, is too small.

**Suitability:**

3

---

### Meta-Review · Area_Chair_1rin · 2024-07-02

**Recommendation:** Accept (Poster)
**Confidence:** 5

**Metareview:**

After rebuttal, all the 4 reviews are positive, so I recommend the acceptance of this paper.